# Mental health interventions for suicide prevention among indigenous adolescents: a systematic review protocol

Antonio José Grande,[1] Christelle Elia ,[2] Clayton Peixoto,[3] Paulo de Tarso Coelho Jardim,[1] Paola Dazzan,[2] Andre Barciela Veras,[1] John Kennedy Cruickshank,[2] Seeromanie Harding[2]

¹State University of Mato Grosso do Sul, Campo Grande, Mato Grosso do Sul, Brazil
²King's College London Faculty of Life Sciences and Medicine, London, UK
³Federal University of Rio de Janeiro, Rio de Janeiro, Brazil

**Correspondence to**
Professor Antonio José Grande; grandeto@gmail.com

## ABSTRACT

**Introduction** There are more than 370 million indigenous people from 5000 cultures living in 90 countries worldwide. Although they make up 5% of the global population, they account for 15% of the extreme poor. Youth suicide is the second leading cause of mortality among 15–29 years old and disproportionately affects indigenous youth. This research protocol pertains to a systematic review of studies that use a comparator/control group to evaluate the effectiveness of suicide interventions targeting indigenous adolescents (aged 10–19 years).

**Methods and analysis** We will conduct a systematic search on MEDLINE, EMBASE, CINAHL, LILACS and PsycINFO from inception to September 2019 to identify articles that compare mental health interventions for suicide prevention among indigenous adolescents. Two reviewers will independently determine the eligibility of each study. Studies will be assessed for methodological quality using the risk of bias tool to assess non-randomised studies of interventions. We will conduct a meta-analysis if possible and use established statistical methods to identify and control for heterogeneity where appropriate.

**Ethics and dissemination** This systematic review will use published data and does not require ethics approval. However, this review is in preparation of a feasibility study that will examine how best to support the physical and mental health of indigenous adolescents in Brazil. Ethics approval for the feasibility study was obtained from National Research Ethics Commission. Findings will be disseminated through a peer-reviewed publication and will be made available to key decision-makers with authority for indigenous health and other relevant stakeholders.

**PROSPERO registration number** CRD42019141754.

## Strengths and limitations of this study

► The systematic review will fill a gap in the evidence base by providing a comprehensive assessment of the existing literature that evaluates the effectiveness of mental health interventions for suicide prevention among indigenous adolescents.

► Rigorous methods of review wll be followed with at least two independent study authors to conduct for screening, data extraction and critical appraisal.

► The findings will inform policy actors and practitioners about feasible interventions to prevent suicide among indigenous adolescents.

► If a paucity of studies is reported then this would locate a research gap that needs to be urgently addressed.

than for non-indigenous populations.[2] A gap in life expectancy at birth (ie, lower in indigenous than non-indigenous populations in the same country) of more than 5 years was recorded for indigenous populations in Australia, Cameroon, Canada (First Nations and Inuit), Greenland, Kenya, New Zealand and Panama. Infant mortality rates for indigenous infants were more than two times than that of those observed for non-indigenous or national populations in Brazil, Colombia, Greenland, Peru, Russia and Venezuela. Poverty, poor education levels, employment status and access to health services are all important contributors to the health disparities. Despite representing a rich diversity of cultures, they continue to be among the world's most disadvantaged groups, regardless of whether they live in high-income countries (eg, the Inuit in Canada) or lower middle-income countries (eg, Baka in Cameroon). The legacy of colonisation and of policies of forced assimilation continue to be a cause of intergenerational trauma, manifested through feelings of marginality, depression,

## INTRODUCTION

There are more the 370 million indigenous people from 5000 cultures living in 90 countries worldwide. Although they make up 5% of the global population, they account for 15% of the extreme poor.[1] Anderson et al conducted a systematic analysis of several social and health indicators for 28 indigenous populations in 23 countries and showed poorer outcomes for indigenous peoples

anxiety and confusion, which places indigenous peoples at increased risk of suicide.[3]

Youth suicide is the second leading cause of mortality among 15–29 years old[4] and disproportionately affects indigenous youth.[5–7] Indigenous children (5–17 years old) in Australia die from suicide at five times the rate of their non-indigenous peers (10.1 per 100 000 vs 2 per 100 000 in 2013–2017). In New Zealand, the suicide rate in Maori youth aged 15–24 years is more than two times than that of non-Maori peers. (40.7 per 100 000 vs 15.6 per 100 000 in 2013–2017). In Canada, the rate among Inuit youth is 11 times than that of the non-indigenous youths.[8] Most interpretations of this gap highlight the persistent social and economic disadvantage experienced by indigenous youth relative to non-indigenous youth.[9] The epidemic of youth suicide is relatively recent in some cultures, with an increase in the latter half of the 20th century, with men accounting for the majority of suicides, and with the 15–24 years age group having the highest suicide rates of any age group.[10 11] Furthermore, suicide among indigenous young people may be unreported due to misclassification. Risk factors include mental health disorders, stressful life events, substance abuse and poor physical health all of which occur at disproportionately higher rates in indigenous populations.[12 13] Suicide in youth is also known to occur in clusters, and suicidal behaviours (eg, ideation, attempts) are strong risk factors for death by suicide.[6] Protective factors include high social support, cultural connectedness and personality factors such as high self-esteem and internal locus of control and increasing age.[6]

Over the last 20 years, indigenous people's rights have been increasingly recognised through international organisations such as the United Nations Permanent Forum on Indigenous Issues, which has a permanent forum for youth.[2] The 2030 Agenda for Sustainable Development refers to indigenous people six times: three times in the political declaration, two times in the target under Goal 2 on Zero Hunger (target 2.3) and Goal 4 on education (target 4.5). Many of the Sustainable Development Goals (SDGs), however, are relevant for indigenous peoples, particularly those with a focus on reducing inequalities and reducing mortality from non-communicable diseases (including suicide) by 33% by 2030. Given the vulnerability of indigenous communities, implementation of the SDGs provide opportunities for policy actors to promote initiatives that improve outcomes among indigenous communities. With such high rates of suicide among indigenous youth,[8] culturally relevant suicide interventions are urgently needed. Many indigenous populations hold a holistic view of health and well-being and interventions need to align with these perspectives and engage with the economic, socioenvironmental and historical issues that contribute to youth suicide in indigenous cultures. There have been two reviews of suicide prevention programmes.[9 10] Both captured studies published up to 2012. Clifford et al[10] reported on two Australian programmes and seven American programmes. The programmes targeted all ages, and there was a general lack of rigorous evaluation designs as only one study included in the review evaluated outcomes using a comparator group. Harlow et al[9] reported on nine evaluations of suicide prevention interventions with youths; five targeted Native Americans; three targeted Aboriginal Australians and one targeted First Nation Canadians. As with the previous review, poor evaluation designs were noted. In recognition of a general lack of methodologically rigorous study designs across geographically and culturally diverse indigenous populations and the need for an updated review, we will use a broad eligibility criterion to maximise the possibility of capturing any study that attempted to evaluate prevention programmes using a comparator group among adolescents.

This article presents the protocol for a systematic review of studies which used a comparator/control group to evaluate the effectiveness of suicide interventions targeting indigenous adolescents (aged 10–19 years).

## OBJECTIVE

To synthesise the scientific evidence on suicide prevention programmes targeting indigenous youth. Our principal research question is: What interventions, including single or multicomponent interventions, prevented suicides (or not) and why did they work (or not)?

## METHODS AND ANALYSIS

This protocol adheres to the Preferred Reporting Items for Systematic reviews and Meta-Analyses Protocols (Equity) guidelines.[14]

### Types of studies

We will include any randomised or non-randomised studies, which has a control or comparative group.

### Types of participants

The participants will be adolescents aged 10–19 years who self-identified as indigenous peoples and are accepted as such by their community.[2] We are guided by the policy definition developed by the International Labour Organization (ILO) in 1989. It characterises indigenous peoples as: tribal peoples in independent countries whose social, cultural and economic conditions distinguish them from other sections of the national community and whose status is regulated wholly or partly by their own customs or traditions or by special laws or regulations; and peoples in independent countries who are regarded as indigenous because of their descent from the populations who inhabited the country, or a geographical region to which the country belongs, at the time of conquest or colonisation or the establishment of present state boundaries and who, irrespective of their legal status, retain some or all of their own social, economic, cultural and political institutions.[15 16]

### Types of interventions

We will include in person or e-health interventions and which have targeted young indigenous people anywhere in the world. We will consider a wide range of delivery channels (eg, in person, online, phone), different practitioners (healthcare practitioners, teachers, lay healthcare providers) and sectors (ie, health, primary, secondary and tertiary care, education, guardianship councils).

### Types of outcome measures

#### Primary outcomes

► Self-injury acts.
► Suicidal ideation.
► Suicide attempts.
► Death by suicide.

#### Secondary outcomes

► Well-being/Quality of life.
► Social Functioning including Educational Outcomes.

### Search methods for identification of studies

#### Electronic searches

We will search the Cochrane Central Register of Controlled Trials (CENTRAL, March 2020, MEDLINE (1966 to March 2020), EMBASE (1974 to March 2020), CINAHL (1981 to March 2020), LILACS (Latin American and Caribbean Health Sciences) (1982 to March 2020) and PsycINFO (1887 to March 2020). We will use the search terms: (Indigenous or Indigenous or native* or Native* or Māori or Maori or Aborigin* or aboriginal or "Torres Strait Island*" or "torres strait island*" or "first nation*" or "first people*" or Inuit or Metis or Métis or ethnic* or "population groups") AND (intervention* or program* or treatment* or treat* or therap* or service* or prevent* or diversion* or initiative*) AND (well-being or "well being" or mental or depress* or anx* or suicide* or trauma* or alcohol* or drinking or cannabis or cocaine or methamphet* or amphet* or substance* or addict* or heal* or empower* or grief or loss* or stress* or psychosis or psychoses or psychotic or resilien* or recovery or "mental health" OR schizophrenia or mania or mood or internalizing or externalizing or affective or behavioural or drugs or "crack cocaine" or addiction or "mental illness" or happiness or emotion* or psych* or psychology) adapted to every other database. We will not use any language restrictions. If articles are not in English, Italian, Lebanese or Portuguese (native languages of the authors), we will use academic networks (eg, Cochrane) to translate the critical parts (methods and results) to enable screening of abstracts and, if included this will be done for the full paper

#### Searching other resources

Experts will be identified via professional organisations (eg, Royal College of Psychiatrists), academic networks and research societies (Social Science and Medicine), medical associations (eg, Brazilian Medical Association) and also via targeted researchers (eg, those at Federal and State universities of Brazil who study indigenous health).

We will search for published and unpublished studies (eg, theses available in electronic format) that may be eligible for inclusion.

### Inclusion criteria

We will include non-randomised and randomised studies, regardless of whether they contained process evaluation although its inclusion in studies will aid the understanding of why an intervention worked or not.

Our primary aim focus is on any intervention that prevents suicide, which may also include studies that used positive psychology and promoted resilience-related competencies, where these were implemented for suicide prevention. These competencies may include the use of coping skills that may ultimately mitigate the emergence of acute states of distress and mental health problems to eventually prevent suicidal behaviour. Improved social–emotional competence can be expected to afford some degree of protection against the development of suicidal ideations and behaviours. Hence, identifying and understanding aspects of interventions (eg, use of problem solving for conflict resolution, community/cultural assets,) that promote resilience are important steps in preventing suicide.[17 18]

Additionally, we aim to capture a broad range of interventions that include either a component or solely targeting suicide prevention in adolescents in the education and community sectors. These may include suicide-specific education programmes, combined suicide-specific education and life skills training programmes, individual-level psychotherapeutic interventions, gatekeeper training, peer/community help, peer gatekeeper training and curriculum-based interventions.

Guided by the ILO definition, studies will be included if participants self-identified as indigenous and were accepted as such by their community According to ILO (1989)[2]; had historical continuity and land occupation before invasion and colonisation; had strong links to territories (land and water) and related natural resources; belonged to distinct social, economic or political systems; had distinct language, culture, religion, ceremonies and beliefs; belonged to non-dominant groups of society with resolution to maintain and reproduce ancestral environments and systems as distinct peoples and communities, and with a tendency to manage their own affairs separate from centralised state authorities.

### Exclusion criteria

If the population included in study is not indigenous or there is no distinction between indigenous and non-indigenous populations.

If youth are not included in study or are only briefly mentioned or adult population.

### Data collection and analysis

Two review authors (AJG, CE) will independently assess all studies identified from the database searches by screening titles and abstracts using EndNote V.X8 software. A third

review author (SH) will resolve any disagreements, and reasons for including and excluding trials were recorded. Next, AJG and CE will independently assess the full-text reports for inclusion against the selection criteria.

After both authors have discussed the results of the selection process and have made a consensus decision on which articles to be included/excluded, the data from each of the articles meeting the eligibility criteria will be extracted.

Qualitative data in the included studies will be reported using a narrative synthesis. Themes reported in the individual studies will be described in a table, including any reported for barriers and facilitators. All the reviewers will initially generate the analytical themes independently, and then collectively as a group so as to minimise bias.

### Data extraction and management

Two review authors (AJG, CE) will independently extract data from the included studies using a standard data extraction form.

A standardised, pre-piloted form will be used to extract data from the included studies for assessment of study quality and evidence synthesis. Missing data will be requested from study authors. The draft format will include:

*Study details*: aim, study design including whether a feasibility study was conducted in collaboration with the community to co-develop the design, design details, country in which study was conducted, details on location of intervention delivery (eg,.city or community) and target condition/risk factor (eg, subthreshold symptoms, experience of child maltreatment).

*Participants:* sample size (intervention and control groups at baseline and follow-up), sociodemographic characteristics (eg, age, gender, ethnicity, socioeconomic status) and attrition from the study.

*Intervention details:* description of intervention including frequency and duration of treatments/sessions, mode of delivery (face to face, internet), format (one to one or group), cultural appropriate content and cost of intervention

*Delivery of the intervention:* setting in which intervention was delivered (school, home, healthcare practice), who delivered the intervention (ie, medical doctor, nurse, psychologist, teacher, lay health worker, peer promotion and so on) and whether it was delivered by one practitioner or a team of individuals or online, fidelity of implementers to protocol, culturally appropriate modes of delivery and whether there was intersectoral collaboration (ie, between health and education or guardianship councils).

The RE-AIM (Reach, Effectiveness, Adoption, Implementation, Maintenance framework) will be used to enhance the assessment of programme elements that can improve the sustainable adoption and implementation of effective, generalised/localised, evidence-based interventions.[19] RE-AIM targets reach of the target population; effectiveness or efficacy of the intervention (impact of an intervention on important outcomes, including potential negative effects, quality of life, and economic outcomes); adoption by target staff, settings or institutions; implementation consistency, costs and adaptions made during delivery and maintenance of intervention effects in individuals and settings over time

### Assessment of risk of bias in included studies

Cognisant of the well-documented limitations of the use of 'western' methods in an indigenous context, our critical appraisal will include identifying culturally appropriate methodologies such as Storytelling and Community-Based Participatory Research, with the inclusion of indigenous peoples in the research process in a way that is respectful and reciprocal. We will include comparator groups as well as randomised study designs in recognition that the former may be more appropriate for the indigenous context.[20 21]

Two review authors (AJG, CE) will independently assess the risk of bias of the included studies using the Risk Of Bias In Non-randomised Studie—of Interventions for non-randomised studies and Risk of Bias tool 2.0 for randomised studies.

### Measures of treatment effect

Types of measurements of treatment effect (outcomes) that may be used:

1. Dichotomous data: we will use risk ratio for likely binary outcomes, prevalence ratio for some outcomes.
2. Continuous data: we will combine the results using the mean difference for measures using the same scale or the standardised mean difference where different scales have been used to evaluate the same outcome.

### Unit of analysis issues

We will consider the individual as the unit of analysis. We will separate the non-randomised from randomised studies if we have enough studies for meta-analysis.

### Dealing with missing data

We will send two emails (one initial, one reminder) to the corresponding author to ask for any missing data or incompletely reported study details. We will check for consistency between studies and analyse each outcome.

### Assessment of heterogeneity

We will assess the inconsistencies between studies using the $I^2$ statistic, which gives the percentage of total variation across studies that is due to heterogeneity rather than chance. We will consider heterogeneity substantial if $I^2$ is over 50%. See below for planned subgroup analysis.

### Assessment of reporting biases

If mismatches are identified between study protocols and reports, we will contact the trial authors to clarify the information. We plan to explore the impact of including such studies by conducting a sensitivity analysis. We will conduct a funnel plot asymmetry test if 10 or more trials are included.

## Data synthesis

We will present the data separately for randomised and non-randomised studies. We will meta-analyse trials if the combination of data on outcomes is possible. We will use a random effects model, independently of the heterogeneity identified. Forest plot graphics produced by RevMan V.5.3 will illustrate the meta-analyses. If the combination of data is not possible, we will present a narrative analysis of individual studies. We will create a 'summary of findings' table for the outcomes and we will present the quality of the body of evidence using the five GRADE (Grading of Recommendations Assessment, Development and Evaluation) assumptions (study limitations, consistency of effect, imprecision, indirectness and publication bias).[22]

## Subgroup analysis and investigation of heterogeneity

We plan to explore the subsets or subgroups of countries/regions, for primary, secondary or tertiary care, type of mental health condition, gender, age, urban/rural area.

## Sensitivity analysis

We will pool included studies to verify whether the impact of risk of bias affects the overall effect. We will explore which studies increased heterogeneity.

## Ethics and dissemination

This systematic review will use published data and does not require ethics approval. However, this review is in preparation of a feasibility study that will examine how best to support the physical and mental health of indigenous adolescents in Brazil. Ethics approval for the feasibility study was obtained from National Research Ethics Commission, protocol CAAE (Certificate of Presentation of Ethical Appreciation): 89604318.0000.8030, analysis #3.100.358, from December of 2018.

Our results will also be presented at national and international conferences and will be made available to key decision-makers with authority for indigenous health.

## Quality of the evidence

Two review authors (AJG, CE) will independently rate the quality of the outcomes. We will use GRADE to rank the quality of the evidence using the Guideline Development Tool software[22] and also the guidelines provided in Chapter 11 of the Cochrane Handbook for Systematic Reviews of Interventions.[23]

## Patient and public involvement

The topic to be covered by the review comes from our empirical experience with the indigenous communities in Brazil and from consultations with researchers in indigenous health, and with practitioners and policy actors responsible for indigenous health. The indigenous people were not directly involved in the writing of this protocol. The results will be disseminated to indigenous communities, practitioners and policy actors to aid the planning and provision of services. The authors are from Brazil and UK and do not self-identify as indigenous.

This review will present the best-available evidence for decision-making on suicide prevention programmes targeting indigenous youth in the literature. It will provide in one single document all primary research on the topic with the quality assessment for each study with a ranking of the quality of evidence.

**Contributors** AJG and CE wrote the protocol and approved the final version of the protocol; CP, PdTCJ, PD, ABV, JKC and SH co-wrote the protocol, critically reviewed and approved the final version.

**Funding** This work was supported by the Medical Research Council (MRC) grant number MR/R022739/1. Fundação de Apoio ao Desenvolvimento do Ensino, Ciência e Tecnologia do Estado de Mato Grosso do Sul, grant number 71/700070/2018.

**Competing interests** None declared.

**Patient and public involvement** Patients and/or the public were not involved in the design, or conduct, or reporting, or dissemination plans of this research.

**Patient consent for publication** Not required.

**Provenance and peer review** Not commissioned; externally peer reviewed.

**ORCID iD**
Christelle Elia http://orcid.org/0000-0002-8298-8440

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
