## [Reviewer comments · BMJ Open]

ARTICLE DETAILS

TITLE (PROVISIONAL)	Mental health interventions for suicide prevention among indigenous adolescents: a systematic review protocol
AUTHORS	Grande, Antonio José; Elia, Christelle; Peixoto, Clayton; Jardim, Paulo de Tarso Coelho; Dazzan, Paola; Veras, Andre Barciela; Cruickshank, John; Harding, Seeromanie

VERSION 1 – REVIEW

REVIEWER	Maree Toombs The University of Queensland
REVIEW RETURNED	18-Oct-2019

GENERAL COMMENTS	This will be a great contribution to the space. I look forward to the review. In Australia we do refer to indigenous with a Capital "I", sounds pedantic but worth your consideration.
--

REVIEWER	Emma Mew Yale School of Public Health, Yale University United States
REVIEW RETURNED	25-Oct-2019

GENERAL COMMENTS	Thank you for the opportunity to review this systematic review protocol. This protocol proposes to collect and potentially meta-analyze outcomes from randomized and non-randomized trials that aimed to evaluate interventions for suicide prevention among Indigenous youth worldwide. The proposed methods appear to be comprehensive, as they use multiple electronic bibliographic databases. Given that many Indigenous communities are prioritizing youth mental wellness and resiliency, I believe the proposed review would fill a much-needed evidence gap. However, this protocol has major methodological limitations, and for this reason, I would not recommend it for publication at BMJOpen. The protocol lacks sufficient detail in terms of systematic review and meta-analytic methods. Please ensure PRISMA-P is followed completely and thoroughly. Some suggestions for revision include: 1. The searches proposed are broad in terms of time frame. You may want to consider a date restriction to improve feasibility.2. How will you determine which parts of the paper are “critical” for translation?3. How will you identify “experts in the field” to contact? Please make this more systematic.4. More explicit and clear detail of inclusion and exclusion criteria, and how this would differ between the title/abstract screening, full-
---

	text screening, and data extraction phases. Please also operationalize some of these constructs to prevent subjectivity in assessment between reviewers. 5. Addition of a definition of Indigenous for the context of this project, ideally in the Introduction. 6. Was the search strategy developed in partnership with a public health/medical librarian? If so, you may consider using the PRESS checklist and procedures (https://www.cadth.ca/resources/finding-evidence/press) to optimize your search strategy. 7. Please provide more detail regarding the screening and extraction phases, and more clarity to distinguish methods applied to title/abstract screening, full text screening, and data extraction components. 8. The protocol states that qualitative data will be collected, but there is no information on how it will be analyzed. Please explain how you will handle qualitative data in your data analysis. 9. Addition of one paragraph stating the self-location of the co-authors (including whether any authors self-identify as Indigenous) and how this would impact study design and interpretation of results. As well, authors could consider commenting on how Indigenous peoples would be consulted in the interpretation of review results. Although ethics approval is not required for a systematic review, given that this is focused on an Indigenous health topic, this protocol could also benefit from considering Indigenous-specific research ethics. Would it be possible to provide explanations as to how the pertinent methodological decisions would be of importance to community (particularly in the selection of outcomes and data extraction items)? As well, would it be possible to re-frame suicide prevention interventions using a more positive perspective (such as re-framing as interventions for resiliency and mental health/wellness) and/or highlight the instances of resiliency and strength in Indigenous communities in your Introduction? Finally, would it be possible to explain how this protocol aligns with best practices in Indigenous systematic review methodology, and discuss limitations where there are discrepancies?
--	---

REVIEWER	Nathaniel Pollock School of Public Health, University of Alberta Canada
REVIEW RETURNED	05-Nov-2019

GENERAL COMMENTS	Thank you for the opportunity to review this protocol for a systematic review of interventions to prevent suicide among Indigenous youth. The authors proposed to synthesize the evidence from randomized and non-randomized studies that evaluated the effectiveness of interventions on suicide-related outcomes with Indigenous youth aged 10-19 years old. Overall, this a succinct and well-written protocol. My comments below reflect my interest seeing the protocol strengthened with some additional clarity, justification, and revisions to the to the scope, rationale, and search strategy. Additionally, minor edits to the language used throughout and the use of references would enhance the writing quality and connection to the literature. Major considerations (1) The type of intervention that is of interest would benefit from some additional clarity. What constitutes a “mental health intervention”? Many interventions, especially in Indigenous contexts
---

	do not originate in mental health systems or services. Based on the review criteria, it appears that interventions in the education and community sectors will be included. I suggest refining the title to reflect this broader scope, and providing a more precise definition in the eligibility criteria. (2) Relatedly, it is also unclear if community, public health, or population level interventions fit in the eligibility criteria. Will they be excluded? Are the authors primarily interested in clinical interventions? In the introduction, it would be helpful if the authors provided some examples of the types of intervention studies that would be eligible for inclusion. This should be based on exploratory searches and would help provide support for the feasibility of the review. There are relatively few experimental studies with Indigenous peoples in general. A review that aims to synthesize studies of suicide-related outcomes among a relatively small subgroup (youth) provides a precise focus, but perhaps risks having a small number of eligible papers. This of course is for the authors to discern. But at this point, some examples would help the reader understand in concrete terms the types of studies that are being sought. (3) The authors also noted that there have been two previous reviews of suicide-related interventions in Indigenous communities. In light of this, I am not clear or fully convinced that the proposed review will add something that has not already been addressed, though am certainly interested if the justification is made. As there are relatively few suicide intervention studies with Indigenous youth, the rationale for this review would be enhanced with an expanded discussion to distinguish this review from previous reviews. (4) The authors have chosen a narrow list of search terms. Several recent reviews in Indigenous health provide comprehensive list of key terms related to Indigenous peoples. Given that the pool of potentially eligible studies for this review is likely small, it would be helpful to be as inclusive as possible with search terms to make sure all studies from all Indigenous populations are identified. (5) In the authors' PROSPERO protocol, the main outcomes are suicidal ideation and suicide attempts. In the present protocol, the authors added two additional outcomes: self injury and 'completed suicide'. Why the difference? Relatedly, the authors have included only 1 key term specific to their outcomes of interest, and did not include MeSH terms. This is a curious exclusion. A recent review of RCTs provides a good benchmark for comparison and to inform the search strategy. Riblet, N. B., Shiner, B., Young-Xu, Y., & Watts, B. V. (2017). Strategies to prevent death by suicide: meta-analysis of randomised controlled trials. The British Journal of Psychiatry, 210(6), 396-402. Relatedly, it is not clear why key terms related to mental illness were included. It would be helpful if a rationale was provided to assess the relevance of these terms. Further, there were no terms related to the intervention types included.
--	--

	The outcomes would benefit from more specific definitions and citations. Minor revisions (6) What tool will be used to assess the risk of bias? (7) Please make sure all statements of fact are referenced throughout. For example, the statement that Indigenous peoples comprise “15% of the extreme poor” should include a citation. (8) The I in “Indigenous” should be capitalized (9) Please integrate additional and more recent sources to bolster strength of the introduction/background and better situate the review in the intervention literature on suicide prevention and Indigenous health. (10) A better source for the following statement could be used: The epidemic of youth suicide is relatively recent in some indigenous cultures, increased over time, more so in the latter half of the 20th century, with men accounting for the majority of suicides, and the 15-24 years old group having the highest suicide rates of any age group [10]. I suggest referencing one or more of the epidemiological studies cited elsewhere in the paper. (11) Avoid statements about ‘committing’ or ‘completing’ suicide – refer to ‘death by suicide’ as per language guidelines (ex. https://www.canada.ca/content/dam/phac-aspc/documents/services/publications/healthy-living/language-matters-safe-communication-suicide-prevention/pub-eng.pdf) (12) The discussion of the SDGs (page 3, lines 53-60) misses an opportunity to talk about reducing mortality from non-communicable diseases (including suicide) by 33% by 2030. I suggest this be addressed briefly. (13) Revise statements that suggest a singular “indigenous culture” as on page 4, line 13. As noted in the intro, cultures are diverse and numerous. (14) Under inclusion criteria, the authors should specify that the bullet points refer to the UN working definition/concept of Indigenous. This should include a citation.
--	--

VERSION 1 – AUTHOR RESPONSE

Reviewer 1

The protocol lacks sufficient detail in terms of systematic review and meta-analytic methods. Please ensure PRISMA-P is followed completely and thoroughly.

The protocol now aligns with PRISMA-P after careful checking.

Added to page 8

We will present the data separately for randomized and non-randomized studies. We will meta-analyse trials if the combination of data for the outcomes is possible. Random-effects models will be used, independent of the heterogeneity identified. Forest plot graphics produced by RevMan 5.3 will illustrate meta-analyses. If the combination of data is not possible, we will present a narrative analysis

of individual studies. We will create a 'Summary of findings' table using the outcomes proposed in this protocol and we will present the quality of the body of evidence using the five GRADE assumptions (study limitations, consistency of effect, imprecision, indirectness and publication bias).

1. The searches proposed are broad in terms of time frame. You may want to consider a date restriction to improve feasibility.

Added to page 5:

We will include all relevant publications of each database from inception up to the month prior to manuscript submission. This will ensure that the review is up to date.

2. How will you determine which parts of the paper are "critical" for translation?

Added to P5:

We will not use any language restrictions. If articles are not in English, Italian, Lebanese or Portuguese (native languages of the authors), we will use academic networks (e.g. Cochrane) to translate the critical parts (methods and results) to enable screening of abstracts and, if included this will be done for the full paper.

3. How will you identify "experts in the field" to contact? Please make this more systematic.

Page 5

Experts will be identified via professional organizations (e.g. Royal College of Psychiatrists), academic networks and research societies (Social Science and Medicine), Brazilian medical association and researchers at Federal and State universities in Brazil who study indigenous health.

4. More explicit and clear detail of inclusion and exclusion criteria, and how this would differ between the title/abstract screening, full-text screening, and data extraction phases. Please also operationalize some of these constructs to prevent subjectivity in assessment between reviewers.

We have elaborated on the inclusion and exclusion criteria (Page 5). We will follow Cochrane methods, with two independent reviewers conducting the screening, reading and extracting of the data and also conducting the critical appraisal of papers included.

5. Addition of a definition of Indigenous for the context of this project, ideally in the Introduction.

We have added a definition of Indigenous that aligns with the ILO (inclusion criteria) (see Page 6).

6. Was the search strategy developed in partnership with a public health/medical librarian?

If so, you may consider using the PRESS checklist and procedures

(<https://www.cadth.ca/resources/finding-evidence/press>) to optimize your search strategy.

We did not enlist the help of a librarian as we have a specialist in systematic reviewer in the team [AG], who has published over 30 Systematic reviews. The search strategy was developed to have good balance of sensitivity and specificity. We improved the search terms which will be adapted across databases (page 6).

7. Please provide more detail regarding the screening and extraction phases, and more clarity to distinguish methods applied to title/abstract screening, full text screening, and data extraction components.

We have now expanded the description of the screening and extraction phases (page 6).

8. The protocol states that qualitative data will be collected, but there is no information on how it will be analyzed. Please explain how you will handle qualitative data in your data analysis.

Added to page 6

If qualitative studies are identified, the themes, particularly in relation to barriers and facilitators, reported in the individual studies will be described in a table. All the reviewers will initially generate the analytical themes independently and then collectively as a group so as to minimize bias.

9. Addition of one paragraph stating the self-location of the co-authors (including whether any authors self-identify as Indigenous) and how this would impact study design and interpretation of results. As well, authors could consider commenting on how Indigenous peoples would be consulted in the interpretation of review results.

Added to page 8. Please also see response to point 11.

10. Although ethics approval is not required for a systematic review, given that this is focused on an Indigenous health topic, this protocol could also benefit from considering Indigenous-specific research ethics.

This is part of wider feasibility study that aims to examine how best to support the physical and mental health of indigenous adolescents in Brazil, and all relevant ethics approvals have been obtained (page 8). We have included the ethics number clearance for the wider study.

11. Would it be possible to provide explanations as to how the pertinent methodological decisions would be of importance to community (particularly in the selection of outcomes and data extraction items)?

The authors are conducting a feasibility study in Brazil to examine whether indigenous community health workers can be embedded in schools in indigenous villages to support physical and mental health of school children. The importance of support for mental health, suicide prevention in particular, arose during partnership building and readiness assessments activities with village leaders, school teachers, school children, parents and health care staff. We are grateful for their insights into an issue they consider to be of high priority. The results will be publicly available to with indigenous communities and health professionals globally via a variety of culturally appropriate dissemination channels and also with academic audiences. The authors are from Brazil and UK and do not self-identify as Indigenous

12. As well, would it be possible to re-frame suicide prevention interventions using a more positive perspective (such as re-framing as interventions for resiliency and mental health/wellness) and/or highlight the instances of resiliency and strength in Indigenous communities in your Introduction? Added to page 6 with relevant references

Our primary aim focus is on any intervention that prevent suicide, which may also include studies that used positive psychology and promoted resilience-related competencies, where these were implemented for suicide prevention. These competencies may include the use of coping skills, that may ultimately mitigate the emergence of acute states of distress and mental health problems to eventually prevent suicidal behaviour. Improved social-emotional competence can be expected to afford some degree of protection against the development of suicidal ideations and behaviours. Hence, identifying and understanding aspects of interventions (e.g. use of problem solving for conflict resolution, community/cultural assets,) that promote resilience is an important step in preventing suicide.

Bjorkenstam C, Kosidou K, Bjorkenstam E. Childhood adversity and risk of suicide: cohort study of 548 721 adolescents and young adults in Sweden. *Bmj-Brit Med J.* 2017;357.

Bachmann S. *Epidemiology of Suicide and the Psychiatric Perspective.* *Int J Env Res Pub He.* 2018;15(7).

13. Finally, would it be possible to explain how this protocol aligns with best practices in Indigenous systematic review methodology, and discuss limitations where there are discrepancies?

Thank you for raising this important point. We are very much aware of the discourse in the published literature on Indigenous research methods and methodologies. We have added to page 7:

Cognizant of the well documented limitations of the use of 'western' methods in an indigenous context, our critical appraisal will include identifying culturally appropriate methodologies such a Community-Based Participatory Research, Storytelling, and inclusion of Indigenous Peoples in the research process in a way that is respectful and reciprocal. We chose to include comparator groups as well as randomised study designs (RCTs) in recognition that the former may be more appropriate for the indigenous context.

Reviewer 2

Thank you for the opportunity to review this protocol for a systematic review of interventions to prevent suicide among Indigenous youth. The authors proposed to synthesize the evidence from randomized and non-randomized studies that evaluated the effectiveness of interventions on suicide-related outcomes with Indigenous youth aged 10-19 years old. Overall, this a succinct and well-written protocol. My comments below reflect my interest seeing the protocol strengthened with some additional clarity, justification, and revisions to the to the scope, rationale, and search strategy. Additionally, minor edits to the language used throughout and the use of references would enhance the writing quality and connection to the literature.

Thank you for the considerations

Major considerations

1. The type of intervention that is of interest would benefit from some additional clarity. What constitutes a "mental health intervention"? Many interventions, especially in Indigenous contexts do not originate in mental health systems or services. Based on the review criteria, it appears that interventions in the education and community sectors will be included. I suggest refining the title to reflect this broader scope, and providing a more precise definition in the eligibility criteria.

Thank you for this comment.

Added to page 8

We aim to capture a broad range of interventions that include either a component, or solely targeting suicide prevention in adolescents in the education and community sectors. These may include suicide-specific education programmes, combined suicide-specific education and life skills training programmes, individual-level psychotherapeutic interventions, gatekeeper training, peer/community-help, peer gatekeeper training, and curriculum-based interventions. These have now been listed among the inclusion criteria and the title has been refined accordingly.

2. Relatedly, it is also unclear if community, public health, or population level interventions fit in the eligibility criteria. Will they be excluded? Are the authors primarily interested in clinical interventions?

We are interested in mental health interventions conducted with indigenous adolescents, whether clinical or non-clinical, but given our focus is on prevention we expect most interventions to be community based or public/population health interventions.

3. In the introduction, it would be helpful if the authors provided some examples of the types of intervention studies that would be eligible for inclusion. This should be based on exploratory searches and would help provide support for the feasibility of the review. There are relatively few experimental studies with Indigenous peoples in general. A review that aims to synthesize studies of suicide-related outcomes among a relatively small subgroup (youth) provides a precise focus, but perhaps risks having a small number of eligible papers. This of course is for the authors to discern. But at this point, some examples would help the reader understand in concrete terms the types of studies that are being sought.

Please see response to point 2 and 4 re description of interventions. Previous reviews highlighted an urgent need for improved methodologies across geographically and culturally diverse Indigenous populations. That was just under a decade ago. Our broad eligibility criterion should identify eligible studies.

4 The authors also noted that there have been two previous reviews of suicide-related interventions in Indigenous communities. In light of this, I am not clear or fully convinced that the proposed review will add something that has not already been addressed, though am certainly interested if the justification is made. As there are relatively few suicide intervention studies with Indigenous youth, the rationale for this review would be enhanced with an expanded discussion to distinguish this review from previous reviews.

Added to page 04

Two reviews (Harlow et al 2014, Clifford et al 2013) included studies up to 2012. With the increasing recognition of the poor health of indigenous peoples globally (e.g. Lancet 2016 issue on Indigenous and Tribal people's health), there is a need to update these reviews. Harlow et al (2014) reported on two Australian programs and seven American programs, but there was a general lack of evaluation. We note that only one study in that review evaluated outcomes using a comparator group. Clifford et al (2013) reported on nine evaluations of suicide prevention interventions; five targeting Native Americans; three targeting Aboriginal Australians; and one First Nation Canadians. All included studies did not use a comparator group for evaluation. Both previous reviews did not assess the methodological quality of the studies, conduct a meta-analysis or evaluate the certainty of the evidence.

5. The authors have chosen a narrow list of search terms. Several recent reviews in Indigenous health provide comprehensive list of key terms related to Indigenous peoples. Given that the pool of potentially eligible studies for this review is likely small, it would be helpful to be as inclusive as possible with search terms to make sure all studies from all Indigenous populations are identified.

Added to page 8 - we have improved the search terms

6. In the authors' PROSPERO protocol, the main outcomes are suicidal ideation and suicide attempts. In the present protocol, the authors added two additional outcomes: self-injury and 'completed suicide'. Why the difference?

We made that change after submitting the protocol, which took about 3 months to be registered. We will amend it as it is important to include these outcomes.

7. Relatedly, the authors have included only 1 key term specific to their outcomes of interest, and did not include MeSH terms. This is a curious exclusion. A recent review of RCTs provides a good benchmark for comparison and to inform the search strategy.

Riblet, N. B., Shiner, B., Young-Xu, Y., & Watts, B. V. (2017). Strategies to prevent death by suicide: meta-analysis of randomised controlled trials. *The British Journal of Psychiatry*, 210(6), 396-402.

We have extended the mesh terms as well as added more synonyms (page 7)

8. Relatedly, it is not clear why key terms related to mental illness were included. It would be helpful if a rationale was provided to assess the relevance of these terms. Further, there were no terms related to the intervention types included. The outcomes would benefit from more specific definitions and citations.

We kept a broad search so as to capture any study that had both an intervention and comparator/control group for evaluation.

Minor revisions

9. What tool will be used to assess the risk of bias?

ROBINS I or ROB 2.0 depending if the study is randomized or non-randomized

10. Please make sure all statements of fact are referenced throughout. For example, the statement that Indigenous peoples comprise “15% of the extreme poor” should include a citation.

Amended as suggested

11. The I in “Indigenous” should be capitalized

Amended as suggested

12. Please integrate additional and more recent sources to bolster strength of the introduction/background and better situate the review in the intervention literature on suicide prevention and Indigenous health.

Amended as suggested

Anderson I, Robson B, Connolly M, Al-Yaman F, Bjertness E, King A, Tynan M, Madden R, Bang A, Coimbra CE Jr, Pesantes MA, Amigo H, Andronov S, Armien B, Obando DA, Axelsson P, Bhatti ZS, Bhutta ZA, Bjerregaard P, Bjertness M B, Briceno-Leon R, Broderstad AR, Bustos P, Chongsuvivatwong V, Chu J, Deji, Gouda J, Harikumar R, Htay TT, Htet AS, Izugbara C, Kamaka M, King M, Kodavanti MR, Lara M, Laxmaiah A, Lema C, Taborda AM, Liabsuetrakul T, Lobanov A, Melhus M, Meshram I, Miranda JJ, Mu TT, Nagalla B, Nimmathota A, Popov AI, Poveda AM, Ram F, Reich H, Santos RV, Sein AA, Shekhar C, Sherpa LY, Skold P, Tano S, Tanywe A, Ugwu C, Ugwu F, Vapattanawong P, Wan X, Welch JR, Yang G, Yang Z, Yap L. Indigenous and tribal peoples' health (The Lancet-Lowitja Institute Global Collaboration): a population study. *Lancet*. 2016 Jul 9;388(10040):131-57.

13 A better source for the following statement could be used:

The epidemic of youth suicide is relatively recent in some indigenous cultures, increased over time, more so in the latter half of the 20th century, with men accounting for the majority of suicides, and the 15-24 years old group having the highest suicide rates of any age group [10].

Many thanks – we have amended as recommended

14. I suggest referencing one or more of the epidemiological studies cited elsewhere in the paper.

Amended as suggested

Bachmann S. Epidemiology of Suicide and the Psychiatric Perspective. *Int J Env Res Pub He*. 2018;15(7).

15. Avoid statements about ‘committing’ or ‘completing’ suicide – refer to ‘death by suicide’ as per language guidelines (ex. <https://www.canada.ca/content/dam/phac-aspc/documents/services/publications/healthy-living/language-matters-safe-communication-suicide-prevention/pub-eng.pdf>)

Thank you for sharing the document. Amended as suggested. We have added a definition of Indigenous that aligns with the ILO (inclusion criteria) page 6

16. The discussion of the SDGs (page 3, lines 53-60) misses an opportunity to talk about reducing mortality from non-communicable diseases (including suicide) by 33% by 2030. I suggest this be addressed briefly.

Amended as suggested, page 3

17. Revise statements that suggest a singular “indigenous culture” as on page 4, line 13.

Amended to acknowledge that indigenous cultures are diverse.

18. Under inclusion criteria, the authors should specify that the bullet points refer to the UN working definition/concept of Indigenous. This should include a citation.

Authors must include a statement in the methods section of the manuscript under the sub-heading 'Patient and Public Involvement'.

Amended as suggested. UN citations included

19. This should provide a brief response to the following questions:
 How was the development of the research question and outcome measures informed by patients' priorities, experience, and preferences?
 How did you involve patients in the design of this study?
 Were patients involved in the recruitment to and conduct of the study?
 How will the results be disseminated to study participants?
 For randomised controlled trials, was the burden of the intervention assessed by patients themselves?
 Patient advisers should also be thanked in the contributorship statement/acknowledgements.
 If there is no patient involved in the study, please state "No patient involved" under the sub-heading 'Patient and public involvement'.
Amended as suggested in the text

VERSION 2 – REVIEW

REVIEWER	Maree tombs The University of Queensland, Australia
REVIEW RETURNED	03-Mar-2020

GENERAL COMMENTS	There are a few sentence structure issues.. Line 29- reword as it does not make sense..not require ethical permission is not required. Other than this, I look forward to the results of this review.
---

REVIEWER	Nathaniel Pollock School of Public Health, University of Alberta, Canada
REVIEW RETURNED	03-Mar-2020

GENERAL COMMENTS	Thank you for the opportunity to re-review this systematic review protocol. Overall, the authors have addressed my previous concerns and feedback. A few additional points to consider that may enhance the quality of the protocol manuscript: (1) The overall objective, "To synthesize the scientific evidence on suicide prevention programmes targeting Indigenous youth" is good but may not be sufficiently specific. A more precise and answerable question may make it easier to assess study eligibility. (2) Under the "Types of Participants", the authors state: "We plan to investigate mental health problems as defined by the DSM-V." What is this for? Mental disorders are not listed as an outcome of interest. Will the review attempt to specifically capture interventions that target Indigenous youth with a mental disorder? This should be removed or clarified. (3) Under "Inclusion criteria", the authors state: "We will also include studies that used data collection techniques (individual interviews, focus groups and observation) or data analysis processes characteristic of the qualitative method (for example: thematic analysis, content analysis)." This is unclear as an eligibility criteria. Given that the review is focused on interventions with comparison groups and quantifiable outcomes, this statement makes it seem as though studies may be included irrespective of whether or not they used quantitative measures. It may help to refine the writing here to clarify. Perhaps the authors only meant that they would examine any
--

	process evaluation information reported in otherwise included studies. (4) Be consistent in use of capitalization of Indigenous
--	---

VERSION 2 – AUTHOR RESPONSE

Reviewer Name

Maree toombs

Institution and Country

The University of Queensland, Australia

Please state any competing interests or state 'None declared':
NIL

Please leave your comments for the authors below

There are a few sentence structure issues..
Line 29- reword as it does not make sense..not require ethical permission is not required.
Other than this, I look forward to the results of this review.

Thank you

This systematic review will use published data and does not require ethics approval

Reviewer: 3

Reviewer Name

Nathaniel Pollock

Institution and Country

School of Public Health, University of Alberta, Canada

Please state any competing interests or state 'None declared':
None

Please leave your comments for the authors below
Thank you for the opportunity to re-review this systematic review protocol. Overall, the authors have addressed my previous concerns and feedback.

A few additional points to consider that may enhance the quality of the protocol manuscript:

(1) The overall objective, "To synthesize the scientific evidence on suicide prevention programmes targeting Indigenous youth" is good but may not be sufficiently specific. A more precise and answerable question may make it easier to assess study eligibility.

Our principal research question is: what interventions, including single or multi-component interventions, prevented suicides (or not) and why did they work (or not)?

(2) Under the “Types of Participants”, the authors state: “We plan to investigate mental health problems as defined by the DSM-V.” What is this for? Mental disorders are not listed as an outcome of interest. Will the review attempt to specifically capture interventions that target Indigenous youth with a mental disorder? This should be removed or clarified.

We deleted the sentence we plan to investigate mental health problems as defined by the DSM-V

(3) Under “Inclusion criteria”, the authors state: “We will also include studies that used data collection techniques (individual interviews, focus groups and observation) or data analysis processes characteristic of the qualitative method (for example: thematic analysis, content analysis).” This is unclear as an eligibility criteria. Given that the review is focused on interventions with comparison groups and quantifiable outcomes, this statement makes it seem as though studies may be included irrespective of whether or not they used quantitative measures. It may help to refine the writing here to clarify. Perhaps the authors only meant that they would examine any process evaluation information reported in otherwise included studies.

Thank you, we meant that we would examine any process evaluation information reported in otherwise included studies

Page 6

We will include non-randomised and randomised studies, regardless of whether they contained process evaluation although its inclusion in studies will aid the understanding of why an intervention worked or not.

(4) Be consistent in use of capitalization of Indigenous

Thank you again. We double checked

The text was re-read and the grammar was properly corrected